# Investigation of Heat Accumulation in Femtosecond Laser Drilling of Carbon Fiber-Reinforced Polymer

**DOI:** 10.3390/mi14050913

**Published:** 2023-04-23

**Authors:** Yaoyao Li, Guangyu He, Hongliang Liu, Mingwei Wang

**Affiliations:** Tianjin Key Laboratory of Micro-scale Optical Information Science and Technology, Institute of Modern Optics, Nankai University, Tianjin 300350, China

**Keywords:** ultrafast laser, carbon fiber-reinforced polymer, ablation threshold, HAZ

## Abstract

Carbon fiber-reinforced polymer (CFRP) has indispensable applications in the aerospace field because of its light weight, corrosion resistance, high specific modulus and high specific strength, but its anisotropy brings great difficulties to precision machining. Delamination and fuzzing, especially the heat-affected zone (HAZ), are the difficulties that traditional processing methods cannot overcome. In this paper, single-pulse and multi-pulse cumulative ablation experiments and drilling of CFRP have been carried out using the characteristics of a femtosecond laser pulse, which can realize precision cold machining. The results show that the ablation threshold is 0.84 J/cm^2^ and the pulse accumulation factor is 0.8855. On this basis, the effects of laser power, scanning speed and scanning mode on the heat-affected zone and drilling taper are further studied, and the underlying mechanism of drilling is analyzed. By optimizing the experimental parameters, we obtained the HAZ < 10 μm, a cylindrical hole with roundness > 0.95 and taper < 5°. The research results confirm that ultrafast laser processing is a feasible and promising method for CFRP precision machining.

## 1. Introduction

Carbon fiber-reinforced polymer (CFRP) is composited with carbon fiber as the fiber reinforcement and epoxy resin as the matrix. It has wide applications in aerospace, the military industry, sports as well as the automotive industry [1,2,3] for its characteristics of low density, high stiffness, excellent corrosion and fatigue resistance. To facilitate the connection of CFRP components, plenty of through holes or blind holes need to be machined. However, its anisotropy and inhomogeneity of bonding between the fibers and matrix provide great difficulties for precision machining. Mechanical machining [4], such as sawing, milling, drilling or grinding, electrical discharge [5] and long-pulse laser machining [6,7], can lead to various forms of damage, such as delamination, fiber pull-outs and a large heat affected zone (HAZ), which decrease the mechanical properties of CFRP [8].

Since the advent of chirped pulse amplification technology, the ultrashort pulsed laser has gradually stepped onto the stage of history, and it has also been applied to precision machining because of its ultrashort pulse width and ultrahigh power density [9]. Wang et al. [10] reviewed the research work on laser drilling of structural ceramics with different pulse widths ranging from millisecond to femtosecond, and the processing quality of the ultrafast laser was found to be significantly better than that of long pulse laser processing. Unlike long pulse laser machining, in which the electrons first absorb the energy of incident photon to reach the excited state, the interaction time between the laser and matter is far shorter than the time the electron transfers energy to the phonon again when using ultrafast laser machining. So, the ultrashort pulse transfers its own energy to the electron in a very short time, and the high power changes the absorption mode of electrons [11,12]. In other words, the heat absorbed from the laser cannot be transferred to the lattice but directly turns the material into a high-temperature, high-pressure and high-density plasma that spurts with heat and processing residues.

It can be said that ultrafast laser processing has fundamentally changed the mechanism of interaction between a laser and materials and become a non-hot-melt or “cold” processing method with ultra-high accuracy, ultra-high spatial resolution and ultra-high throughput [13], thus opening a new field of laser processing. It also provides a reliable solution for laser precision processing of CFRP [14], such as micromachining [15], surface modification [16] and laser-induced periodic surface structures [17].

Defects generated during machining such as the HAZ, charring, matrix recession and delamination are major obstacles in advancing laser machining of CFRP [18,19], which are mainly attributed to the interaction between the laser beam and at least two different materials with large material performance differences. Figure 1a shows the internal carbon fiber bundles in CFRP with surface damage under an optical microscope. CFRP is made up of a single-layer carbon fiber cloth with a certain thickness layer by layer. The laying angle and the number of layers of the single-layer carbon fiber cloth vary for different performance requirements, as shown in Figure 1b. Among these defects, the HAZ is responsible for the decrease in the strength performance of the composite materials [20]. Consequently, the HAZ is considered the major obstacle to wide industry applications of laser machining of CFRP. Freitag et al. [21] first reported the fact that the fraction of heat remaining in the CFRP after each ablation process depended on both the pulse repetition rate and the pulse energy. J. Finger et al. [20] realized the HAZ of the cutting edge to be smaller than 5 μm and precise cutting of 2 mm thick CFRP samples with a picosecond laser. Zhang et al. [22] processed CFRP with different pulse widths and measured HAZs on the material surface, which showed that the HAZ cutting with an ultrafast laser is much smaller, and the edge is smooth almost without processing defects. Choi et al. [23] found that picosecond laser machining of CFRP has more advantages both in processing cone angle and in tensile properties. Wang et al. [24] used the picosecond laser-induced plasma microdrill to process CFRP and proved that better micro hole roundness and quality could be obtained than that in long pulse laser machining. Shen et al. [25] found that the increased surface slope and the formation of the highly reflective recast layer were the decisive factors in the laser–carbon interaction, which affected the efficient absorption coefficient of the laser and resulted in the nonlinear drilling rate and the self-limiting of the drilling.

In addition, the processing parameters and methods [26,27] are also influential. Fujita et al. [28] explored changes in the HAZ and cutting efficiency of CFRP at different laser wavelengths ranging from 266 to 1064 nm and confirmed that shorter laser wavelengths could achieve smaller HAZs and higher cutting efficiency. Herrmann et al. [29] compared the processing results of three different beam delivery systems, such as the galvanometer scanner, trepanning head and diffractive optical elements to drilling processes of CFRP. Ouyang et al. [30] used the “double rotation” cutting technique to drill holes on the CFRP surface using a picosecond laser and found that greater slit width and higher drilling efficiency can be achieved. Li et al. [31] proposed a special interlaced scanning mode which effectively reduced the heat accumulation effect of adjacent laser beam scanning paths and better removed the material. Jia et al. [32] summarized hybrid laser processing technologies, which combined lasers with different pulse widths to achieve high-efficiency processing. The results showed that the heating of a long-pulse laser can accelerate the expansion of plasma and reduce the energy lost by the plasma shielding. The millisecond laser pulse can increase the laser absorption or improve the sputtering of the melt [33], which proves that combined laser technology is a reliable processing method.

Although many processing methods have been proposed, there are still technical bottlenecks in the ultrafast laser processing of CFRP to restrain defects and improve processing efficiency. How to further improve the cutting quality and machining efficiency is worth exploring. In this paper, the ablation threshold of CFRP and the influence of machining parameters on the surface topography were determined. In addition, an alternate scanning drilling method is proposed to reduce the accumulation of thermal damage during machining. How to pursue the “cold” machining parameter window as much as possible is a major challenge and research hotspot in the field of ultrafast laser machining of CFRP, and the new technology scheme to improve machining efficiency and large size structure machining under the premise of ensuring precision machining quality is also the focus of this paper.

## 2. Materials and Methods

The CFRP samples used were purchased from Jiangsu Boshi Carbon Fiber Co., Ltd. (Nanjing, China) and the specifications is shown in Table 1. According to the component requirements, the prepregs are cut and sealed in a vacuum bag in the mold. After heating, pressurization, insulation (medium- or high-temperature), cooling and pressure relief processes, they become the required shape and quality forming. Based on its strength and tensile modulus, CFRP can be divided into various series. Considering the scope of application, CFRP of the T700 level was selected.

The femtosecond laser machining experimental setup and the drilling method are shown in Figure 2. Because the laser spot size is too small relative to the thickness of the CFRP plate, the single-layer scanning at the fixed focal point cannot realize the through-hole machining. Therefore, a concentric circular orbit is completed at a certain scanning distance to generate annular grooves with a certain depth. Then, the laser focus is controlled to drop a certain distance, and the same concentric circle trajectory is scanned in the next layer until it drops to the underside of the material, completing the through-hole machining of the CFRP plate, as shown in Figure 2b. An f-θ objective with a focal length of 10 mm is used to focus the laser beam of the amplified Ti-sapphire femtosecond laser system (Astrella, Coherent Inc., Santa Clara, CA, USA). The system can deliver 1 kHz linear polarized pulses with the pulse width changing from 35 fs to 150 fs, and an energy of 6 mJ at a central wavelength of 800 nm. The accurate adjustment of incident energy is achieved with an electric attenuator. The experimental samples are placed on the 6-axis maneuvering platform to realize the specific machining design. Finally, the micromachining pulse width is set to 90 fs as shown in Table 2.

In order to observe the surface morphology of the processed samples, the OLYMPUS BX51 microscope is used to characterize and analyze the surface morphology of the processed areas. Defects such as delamination, charring and fiber pull-outs can be observed, and the width of the HAZ can be measured under the magnification of the optical microscope. In particular, the three-dimensional optical profiler (NewView 9000, Zygo Co., Middlefield, CT, USA) can characterize the processing zone to sub-nanometer accuracy. In order to guarantee the accuracy of the results, all experimental data were measured multiple times before taking the average value. All measurements and morphology analysis were performed at room temperature (~25 °C). Before the experiment, the samples were soaked in ethanol and washed for three minutes with an ultrasonic cleaner to remove surface stains.

In order to measure the micro-hole geometry and evaluate the drilling quality, the least square method and the least area method were used to calculate the diameter and roundness errors of the inlet and outlet. The method for measuring the roundness of the micro-hole is shown in Figure 3a. There are at least two contact points between two concentric circles and the hole wall, and the radius difference in the concentric circles ΔR is defined as the roundness error. The taper statistics are shown in Figure 3b, and the calculation formula is as follows:$$T=arctan\frac{w_{t}-w_{b}}{2d}$$

## 3. Results and Discussion

### 3.1. Determination of the Ablation Threshold

The absolute calibration that the effect of laser energy has on materials is a necessary condition for the quantitative study of the interaction between a laser and materials. As an important parameter in ultrafast laser processing, the ablation threshold can not only be used to control the laser energy to achieve different processing effects but also is a key to retrieving the laser–material interaction mechanism from the processing phenomenon.

Therefore, it is necessary to measure the ablation threshold of CFRP. In general, the ablation threshold can be measured with direct observation and laser emission spectrum [34]. Both direct observation and laser emission spectrum need real-time observation in the experiment. Therefore, a more convenient and widely used numerical calculation method is adopted. Using the relationship between ablation area or volume and laser pulse energy density, the linear regression equation is obtained by fitting the experimental data with the least square method, and the ablation threshold is calculated by setting the area or volume to zero [35,36,37,38].

The single-pulse energy of the femtosecond laser obeys a Gaussian distribution, and the relationship between laser energy density and beam section radius is as follows:$$E_{p}=\int_{0}^{+\infty }2\pi r\varphi \left(r\right)dr=\int_{0}^{+\infty }\varphi_{0}exp\left(\frac{-2r^{2}}{\omega_{0}^{2}}\right)2\pi rdr=\frac{\pi \omega_{0}^{2}}{2}\varphi_{0}$$
where $\varphi_{0}$ is the peak energy density of the laser beam; *r* is the distance to the beam center; and $\omega_{0}$ is the radius of the laser spot. Assuming that the diameter of the effective ablation area of the single-laser pulse is *D*, and the energy density of the laser beam at the edge of the effective ablation area is $\varphi_{h}$, using the relationship between the energy of the single laser pulse and the energy density of the center, we obtain:$$D^{2}=2\omega_{0}^{2}\ln\frac{\varphi_{0}}{\varphi_{h}}=2\omega_{0}^{2}(\ln P_{avg}+\ln\frac{2}{\pi f\omega_{0}^{2}\varphi_{h}})$$

The trend in the femtosecond pulse ablation threshold with the number of effective pulses can be expressed with the pulse accumulation factor [39]:$$N=\frac{2f\omega \left(z\right)}{v},$$
$$\varphi_{h}\left(N\right)=\varphi_{h}\left(1\right)N^{S-1},$$
where *N* is the number of effective pulses, f is the repetition frequency, $\omega \left(z\right)$ is the laser spot radius, v is the scanning speed during processing, $\varphi_{h}\left(N\right)\;$ is the ablation threshold for *N* effective pulses, $\varphi_{h}\left(1\right)$ is the ablation threshold in the case of a single pulse and *S* is the pulse accumulation factor.

Figure 4a shows the surface morphology of the cleaned sample, while Figure 4b shows femtosecond laser cutting on the CFRP surface. The multi-pulse ablation threshold can be calculated by scribing the sample and measuring the ablation diameter at different energy and scanning speeds. Thus, the single-pulse ablation threshold can be obtained with the pulse accumulation factor. The simulation results are presented in Figure 5.

Table 3 lists the corresponding ablation threshold, effective pulse number and corresponding simulated spot radius at different scanning speeds. Obviously, the ablation threshold gradually decreases with the increase in scanning speed, which is basically consistent with the fitting in Figure 5d, in which *a* represents the single-pulse ablation threshold of CFRP and *b* represents *S* − 1. The reduction in scanning speed means that the longer the laser action time per unit area, the more effective pulses per unit area of laser action, and, therefore, the more energy deposited on the material surface. It can be seen from Figure 5d that the ablation threshold of CFRP decreases with the increase in the number of incident laser pulses, and the rate of the threshold decreases gradually, indicating that the threshold will reach saturation when the number of pulses increases to a certain extent. This “incubation” effect is caused by the accumulation of laser-induced defects in the material, which is the result of multiple-laser pulse irradiation. In the process of multi-pulse laser ablation, there is a situation that the previous pulse is completed, and the energy is not dissipated. While the next pulse is incident and leads to the energy accumulation effect in the processing, resulting in the continuous increase in the lattice temperature of the material. Therefore, under the energy accumulation effect, the effective ablation threshold of the material that has not been removed decreases, and the material removal rate increases. When the number of laser pulses exceeds a certain number, the material’s absorption of laser pulses approaches saturation, and the ablation threshold finally becomes stable. According to the formulas, the pulse accumulation factor and single-pulse ablation threshold can be calculated as 0.8855 and 0.84 J/cm^2^, respectively, which can provide some guidance for the next processing scheme. In addition, the laser spot radius used in the experiment is 3 μm. Due to the large surface roughness of CFRP and the large thickness tolerance of the material, the different positions of the material surface are not exactly located at the same position in the Gaussian laser beam waist, resulting in a certain error in the simulated spot radius.

To sum up, changing the scanning speed or changing the laser power essentially changes the amount of energy injected into each unit area of the material per unit time, which determines the damage form of the laser to the material. However, the damage form will change with the performance and state of the material itself, which influence the incident laser energy conversely.

### 3.2. The Drilling Effect of Laser Machining Parameters

First, combined with the CFRP ablation threshold, the influence of laser energy on material drilling was explored to find the best processing parameters. After preliminary experiments, different laser powers were used on a 0.3 mm thick CFRP sample at a scanning speed of 0.5 mm/s and a scanning space interval of 10 μm. Because the machining aperture is only a few hundred microns, when the laser energy is too large, the surface will be seriously ablated. Meanwhile, when the laser energy is very low, it is difficult to complete the drilling even though there is almost no heat accumulation on the surface, as shown in Figure 6a. This is because the laser pulse energy will be absorbed by the sidewall and cause thermal damage with the increase in the cutting depth. In addition, the plasma plume of the laser beam, the accumulated processing residues and the scattering of molten debris will cause beam divergence. As shown in Figure 6g, the inlet and outlet diameters are significantly different, and the outlet diameter is more unstable due to multiple interference factors when the laser is downward.

We define energy flux density ED to combine the effects of the average laser power, scanning speed and spot diameter to better examine the effects of laser parameters:$$D=\frac{P_{average}}{v_{scan}\varphi_{0}}$$
where $P_{average}$ is the average laser power, $v_{scan}$ is the scanning speed used in machining and $\varphi_{0}$ is the spot diameter of the laser beam.

The results show that when the laser energy is small, the surface morphology of CFRP is good, and there is almost no obvious HAZ. However, with the increase in laser energy, the heat accumulates gradually. The inlet diameter increases slowly, and eventually approaches stability, as shown in Figure 6g. The outlet diameter is affected by many factors, such as the shielding of the residue and the divergence of the beam, but its overall trend is increasing. As shown in Figure 7a, the width of the HAZ decreases first because the ablation of the material is not complete at low laser energy. Low laser energy makes it difficult to scan the material downward, which not only reduces the processing efficiency but also requires multiple scanning in order to fully remove the material, resulting in a large HAZ. As the average laser power increases, the single-pulse energy also increases gradually. At this time, most of the energy is used for the complete removal of materials, the HAZ is small and the energy utilization rate is high. When the most appropriate processing power range is reached, the HAZ of CFRP also reaches a minimum value. With the further increase in laser power, the HAZ increases sharply because the injected energy is greater than the energy required for material removal, and the excess energy will accumulate and lead to an expansion of the HAZ. If the incident energy flux and pulse number continue to increase, more defects such as edge collapse, delamination damage and burning phenomena will gradually appear during processing. At this time, not only the processing energy is too large a waste, but also the processing edge quality is poor and low precision.

It is difficult to avoid the deviation using the vertical top-down laser focusing scheme, which will lead to an increase in the taper and a decrease in the roundness and affect the quality of laser drilling, as shown in Figure 8. In the process of the laser interaction with CFRP, plasma and debris are often generated, resulting in a lot of scattering and reflection in the laser’s downward propagation path. In addition, the defects caused in the early stage, such as the delayed untimely removal of edge material, will directly affect the propagation path and light intensity distribution in the subsequent laser pulses, resulting in a significant difference between the inlet and outlet diameters.

In addition, the effects of different scanning speeds on the hole morphologies were studied at P = 7.57 mW.

When low-speed cutting is used, more energy can be injected per unit of time. However, this also leads to more accumulated energy and the formation of the HAZ. With the increase in scanning speed, the energy density of the material surface gradually decreases, as shown in Figure 9, which means that the accumulated energy during drilling is less, so the HAZ is significantly reduced. However, when the speed is too high and the energy acting on the material surface is too low, the surface morphology of micropores will be greatly affected because drilling is not formed in one go. On the one hand, too low energy is not enough to realize the complete removal of the material; on the other hand, because the laser beam itself will be defocused or diverged during the deep drilling process, and the side wall will absorb the energy, it will be more difficult to aggravate the laser downward processing. Therefore, when the speed is too high or too low, not only the processing quality is greatly affected but also the processing efficiency cannot be guaranteed. Combining the roundness and taper of the hole, we can see that a satisfactory drilling hole can be achieved at the scanning speed of 0.5 mm/s, as shown in Figure 10 and Figure 11.

### 3.3. The Drilling Effect of the Laser Scanning Method

Experimental studies [40] show that the ablation rate of the material will gradually decrease with an increase in the material thickness until the energy density absorbed by the material is below the ablation threshold. Finally, when the ablation rate drops to zero, it results in a cone. Possible reasons for the reduction in the ablation rate are mainly the influence of laser energy coupling and laser-induced plasma in the hole. In order to ensure the successful fabrication of micropores, the laser power is usually set greater than the ablation threshold of the material, but the top-down processing method can cause serious thermal ablation.

At present, the thermal effect in ultrashort pulse laser processing materials still exists, which mainly comes from the heat accumulation effect. Using high-frequency and low-speed processing parameters will lead to an increase in the overlap rate of light spots, which means an increase in the number of effective pulses, leading to a reduction in the time interval between adjacent pulses and a significant thermal accumulation effect. Therefore, an interlaced scanning method is proposed to improve the surface quality of micro-holes, as shown in Figure 12.

In this ling of a thick CFRP plate, when the laser is focused on the upper surface of the material, an annular groove with a certain depth was generated by completing the concentric circle track with a certain scanning distance. Then, the laser focus was controlled to drop a certain distance (less than the depth of focus), and the same concentric circle track was scanned in the next pass until it dropped to the bottom surface of the material to complete through-hole machining of the CFRP plate. In the concentric circle filling scanning of any pass, as shown in Figure 12a, the default scanning mode of the existing laser beam is the sequential scanning mode, performed from the outside to the inside. The interlaced scanning mode is obtained by changing the scanning sequence shown in Figure 12b. Except for the scanning sequence, the scanning trajectory in the two modes are exactly the same. By changing the scanning spacing *h* between the concentric circles, the influence of the light spot overlap rate on the processing can also be explored.

Since the radius of the laser spot radius used in the experiment is 3 μm, h = 3, 5, 8 and 10 μm were, respectively, used for the experiment, and the results are shown in Figure 13. From Figure 14, it can be seen more obviously that the interlaced scanning mode can effectively reduce the thermal accumulation effect and surface melting in adjacent laser beam scanning paths. What is more, the sidewall structure is much smoother.

Compared with the sequential scanning method, the interlaced scanning method can effectively reduce the HAZ on the material surface, as shown in Figure 15. When h = 3 μm, the laser spot overlap rate of the material is high, which means the laser will continue to irradiate other paths during the scanning process. The heat accumulation effect will cause serious ablation on the surface of the CFRP. At the same time, the broken fibers and steam can be discharged in time during the drilling to reduce the thermal decomposition in the base material at the cut edge and better remove the material. Moreover, the interlaced scanning mode can provide enough time for the volatilization and cooling of materials. When the scanning is too close, the HAZ increases, and the material cannot be completely removed. The remaining debris and fibers will prevent the laser from processing further down, and the sidewall of the drilled hole will have a large taper.

After the experiments, the best combination of process parameters and scanning mode was finally adopted, and the obtained surface morphology is shown in Figure 16. The surface HAZ can be reduced to 10 μm with a small taper and roundness greater than 0.95. According to the morphology obtained with the three-dimensional optical profiler, the side wall is obviously smooth.

## 4. Conclusions

This paper explores the feasibility of femtosecond laser precision machining of CFRP. The single-pulse ablation threshold and multi-pulse cumulative ablation threshold of CFRP are determined using different ultra-fast laser energy densities to conduct single-pulse ablation and multi-pulse cumulative ablation experiments. The interaction mechanism and energy accumulation effect of material under the effective pulse number are analyzed. By optimizing the laser processing parameters and scanning mode, the laser precision processing parameter window and the interlaced scanning mode are determined. This reduces the heat accumulation during the drilling and scanning process and makes the HAZ less than 10 um with no cracks, no metamorphic layer or other defects and a hole roundness >0.95. This research proves the feasibility of femtosecond laser precision processing of CFRP and is of great significance for efficient precision processing of CFRP to meet the needs of the aerospace industry.

## Figures and Tables

**Figure 1 micromachines-14-00913-f001:**
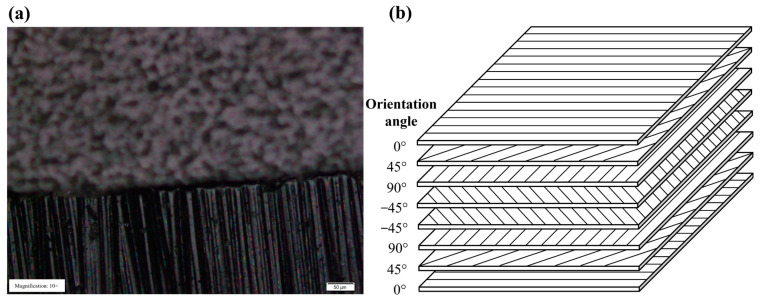
(**a**) CFRP samples and (**b**) CFRP laminates.

**Figure 2 micromachines-14-00913-f002:**
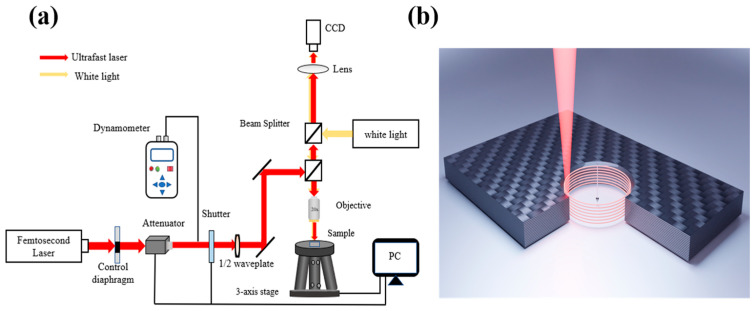
(**a**) Schematic diagram showing the experimental setup. (**b**) Schematic diagram showing laser drilling.

**Figure 3 micromachines-14-00913-f003:**
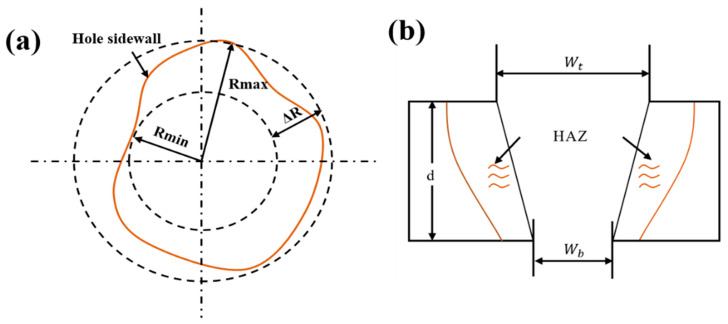
Schematic diagram showing the minimum zone method and taper method. (**a**) Minimum zone method. (**b**) Taper method.

**Figure 4 micromachines-14-00913-f004:**
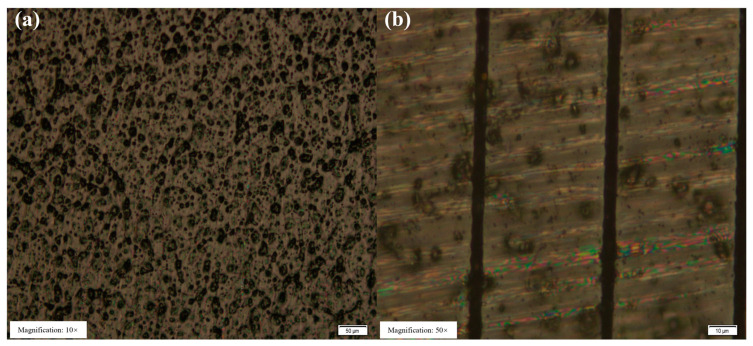
(**a**) Sample surface morphology obtained with the OLYMPUS BX51 microscope. (**b**) Lineation of the sample surface (P = 90 μW, v = 0.1 mm/s).

**Figure 5 micromachines-14-00913-f005:**
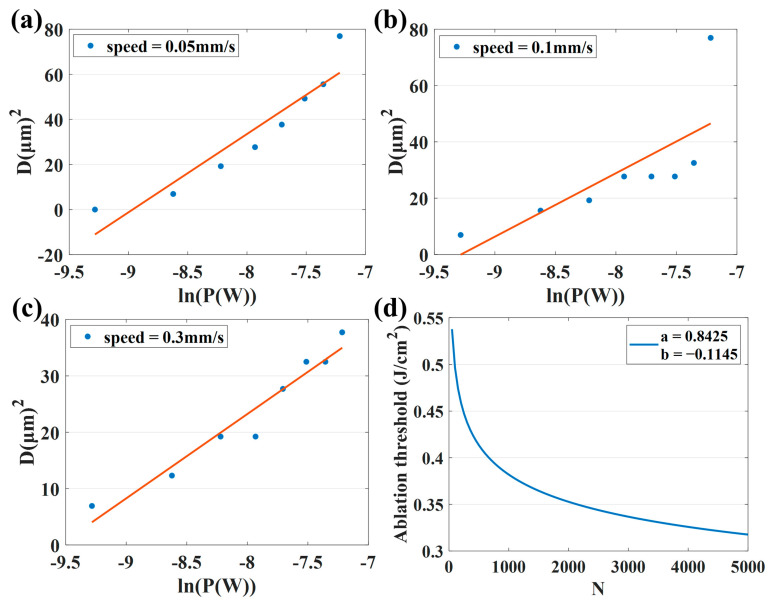
Ablation threshold fitting diagram obtained at different effective pulse numbers: (**a**) v = 0.05 mm/s; (**b**) v = 0.1 mm/s and (**c**) v = 0.3 mm/s. (**d**) Pulse accumulation factor fitting diagram.

**Figure 6 micromachines-14-00913-f006:**
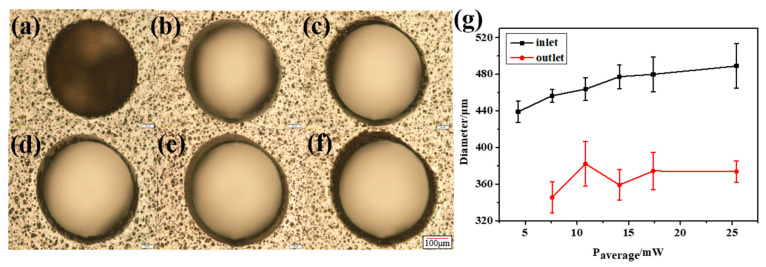
Micro-holes at different laser energies. (**a**) P = 4.32 mW; (**b**) P = 7.6 mW; (**c**) P = 10.83 mW; (**d**) P = 14.11 mW; (**e**) P = 17.38 mW and (**f**) P = 25.42 mW; Scale: 50 μm. (**g**) Inlet and outlet diameters.

**Figure 7 micromachines-14-00913-f007:**
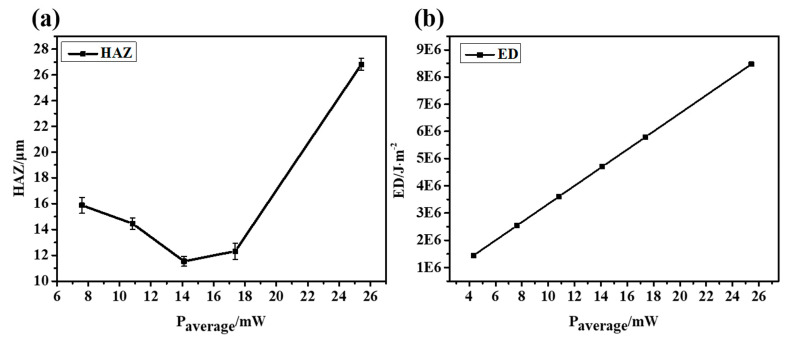
The HAZ and ED at different average laser powers. (**a**) HAZ and (**b**) ED.

**Figure 8 micromachines-14-00913-f008:**
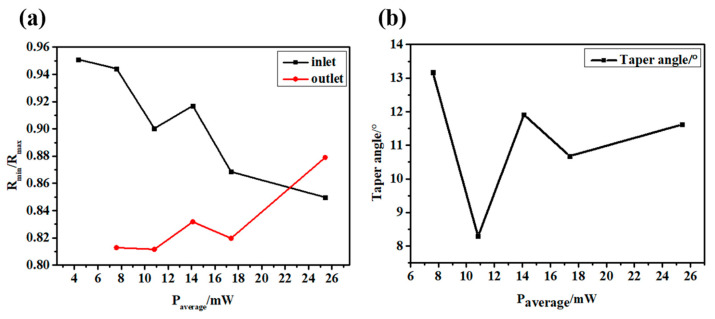
(**a**) Inlet and outlet roundness at different laser powers. (**b**) Taper angle at different laser powers.

**Figure 9 micromachines-14-00913-f009:**
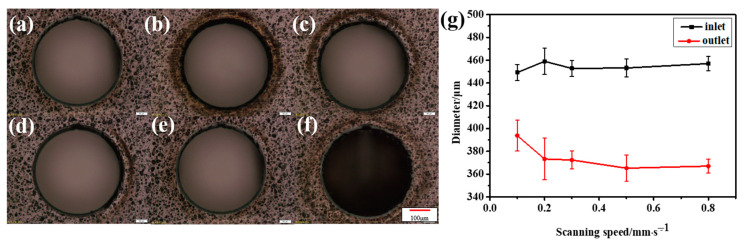
Micro-holes at different scanning speeds. (**a**) v = 0.1 mm/s; (**b**) v = 0.2 mm/s; (**c**) v = 0.3 mm/s; (**d**) v = 0.5 mm/s; (**e**) v = 0.8 mm/s and (**f**) v = 1 mm/s. Scale: 50 μm. (**g**) Inlet and outlet diameters.

**Figure 10 micromachines-14-00913-f010:**
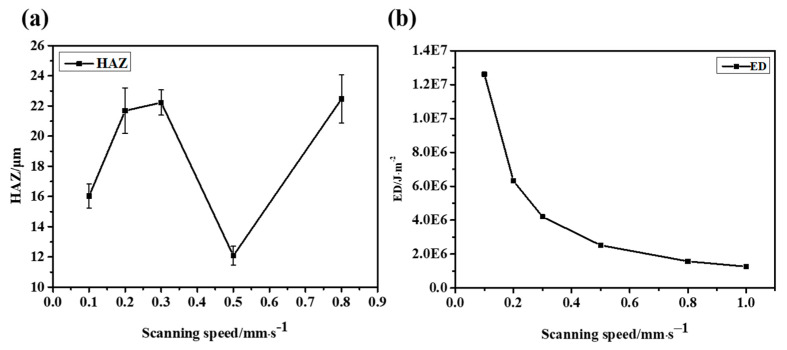
**The** HAZ at different scanning speeds. (**a**) HAZ and (**b**) ED.

**Figure 11 micromachines-14-00913-f011:**
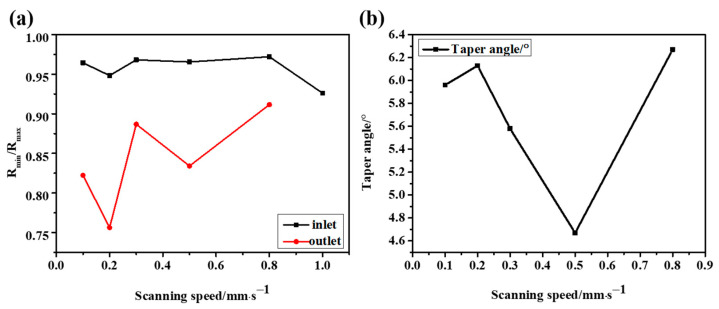
(**a**) Inlet and outlet roundness at different scanning speeds. (**b**) Taper angle at different scanning speeds.

**Figure 12 micromachines-14-00913-f012:**
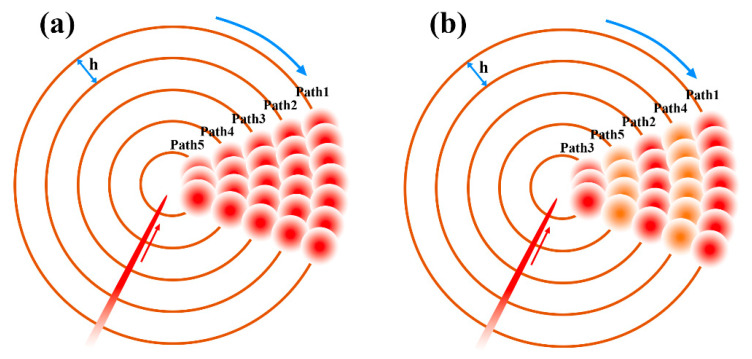
(**a**) Sequential scanning method. (**b**) Interlaced scanning method.

**Figure 13 micromachines-14-00913-f013:**
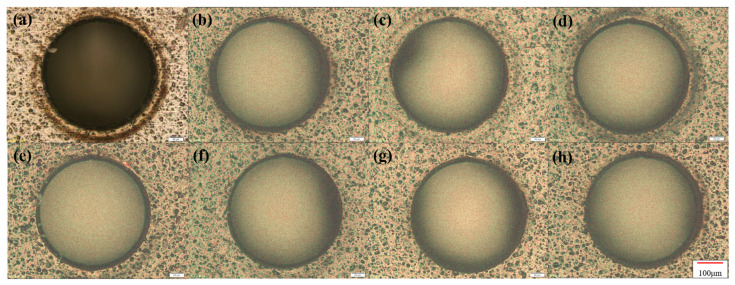
The sequential scanning method: (**a**) h = 3 μm; (**b**) h = 5 μm; (**c**) h = 8 μm and (**d**) h = 10 μm. The interlaced scanning method: (**e**) h = 3 μm; (**f**) h = 5 μm; (**g**) h = 8 μm and (**f**) h = 10 μm. Scale: 50 μm.

**Figure 14 micromachines-14-00913-f014:**
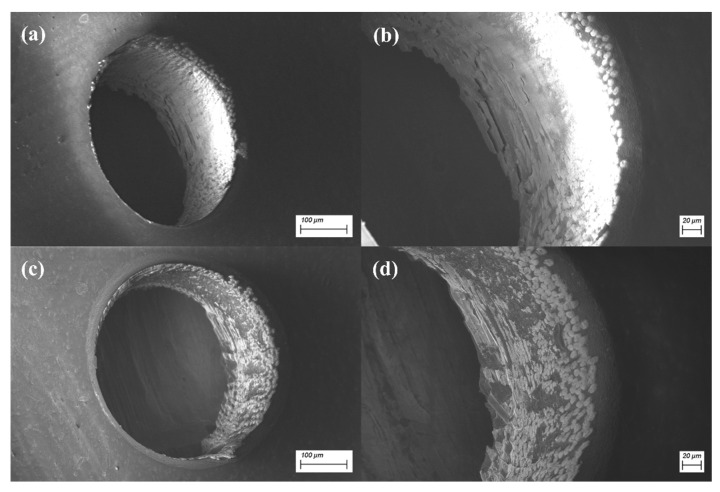
(**a**) SEM image showing the interlaced scanning method. (**b**) Side wall of the interlaced scanning method. (**c**) SEM image showing the sequential scanning method. (**d**) Side wall of the sequential scanning method.

**Figure 15 micromachines-14-00913-f015:**
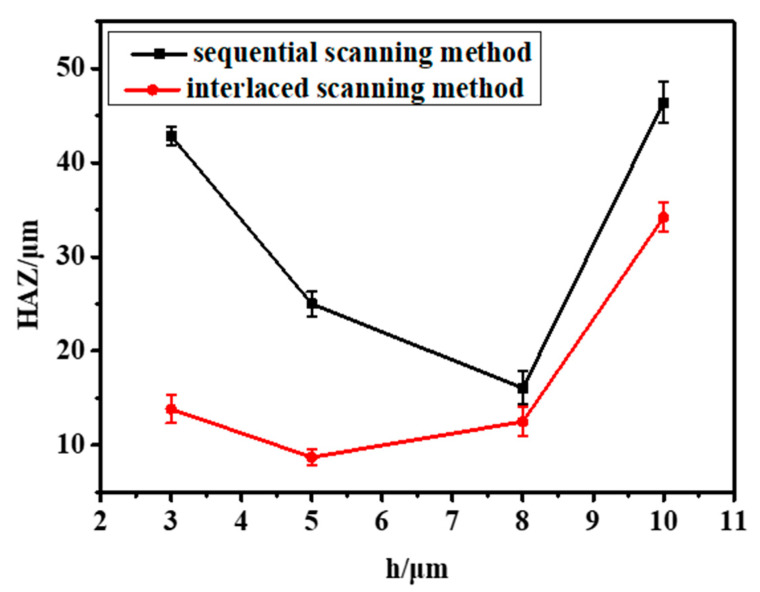
The HAZ at different spacing distances.

**Figure 16 micromachines-14-00913-f016:**
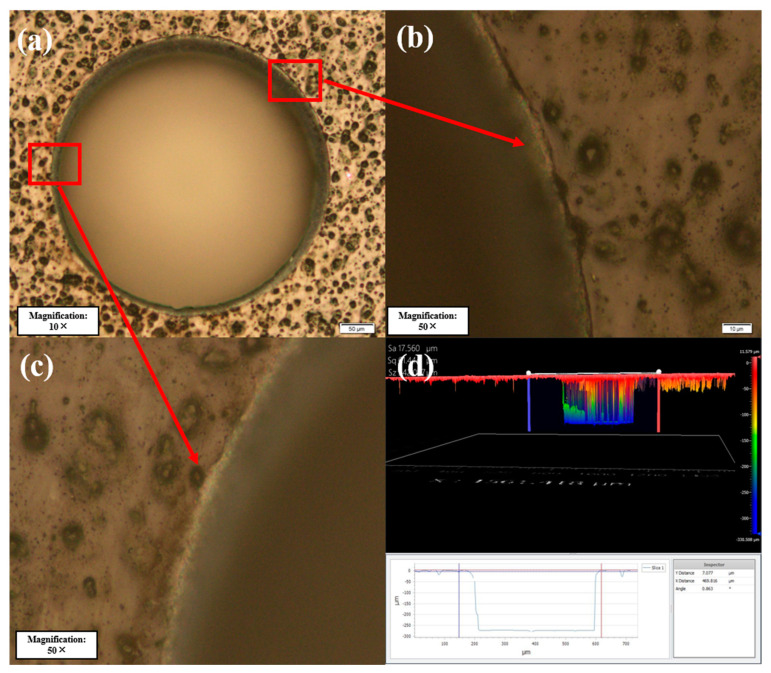
(**a**) Drilling of CFRP. (**b**,**c**) Morphology of the HAZ at the edge of the hole. (**d**) Morphology obtained using the three-dimensional optical profiler.

**Table 1 micromachines-14-00913-t001:** Specifications of CFRP.

Components	Based resin	YB01
	Carbon fiber	T700
Laminate Mechanical Properties	0° Tensile strength	2300 MPa
	0° Tensile modulus	115 GPa
	0° Flexural strength	1250 MPa
	0° Compressive strength	1050 MPa
	Interlaminar shear strength	55 MPa
	size	2 cm × 2 cm × 0.3 cm

**Table 2 micromachines-14-00913-t002:** Specifications of the experimental system.

Varied Parameters	Range
Average Power	0–5 W
Repetition rate	1 kHz
Scanning speed	0–95 mm/s
Pulse duration	90 fs
Wavelength	800 nm
Objective lens focal length	10 mm
Working distance of objective lens	20.5 mm

**Table 3 micromachines-14-00913-t003:** Ablation thresholds at different scanning speeds.

Scanning Speed (mm/s)	0.5	0.4	0.3	0.2	0.1	0.05
$\varphi_{h}$ (J/cm^2^)	0.6684	0.6072	0.6042	0.5788	0.5266	0.4685
*N*	12	12	18	22	67	166
$\omega_{0}$ (μm)	3.1073	2.5596	2.7357	2.1662	3.3580	4.16

## Data Availability

Data available in a publicly accessible repository.

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
