# Peer review of "Investigation of Heat Accumulation in Femtosecond Laser Drilling of Carbon Fiber-Reinforced Polymer"

_micromachines, 2023, doi:10.3390/mi14050913_

Round 1

Reviewer 1 Report

Dear authors

The manuscript is interesting and useful for researchers. Please consider following major comments:

1-Please add description of Figure 1. into the main text and in related section. It was not referenced into the main text.

2-Please add complete description of Figure 2. for better reader understanding.

3-In line 150, please correct the sentence related to Figure 4. Also, please mention the analysis method used for determining morphology in this sentence and correspond description below the picture.

4-In materials and method section, please explain how did fabricated the CFRP samples. It means which kind of materials (such as polymer and carbon fiber) were used and explain their suppling sources. Also, please add complete description of composition details of fabricated CFRP. Please mention the type of CFRP used in this work in abstract, introduction, and conclusion sections.

You just explained description of used setup for machining of CFRP sample.  

5-In Figs. 6, 10, 14, and 16, please correct description of the pictures by adding used analysis method.

5-There are numerous spelling and grammatical errors that must be corrected.

Sincerely

Author Response

Thank you for your comments and please see the attachment.

Reviewer 2 Report

The authors report a review paper on the femtosecond laser drilling of Carbon Fiber Reinforced Polymer. Recently, CFRP as a typical hard and brittle material, its micro-hole processing is indeed a key concern in the field of laser processing. In my opinion, the topic of this paper is of interest, both scientific and industrial. However, the manuscript still needs to be revised before it is accepted for final publication. The authors should consider the following comments/questions.

(1)    The authors need to further polish the language. There are some long and complex sentences in this paper, which may make the readers confused. Please avoid this kind of complex expressions in English for the reader' easy and clear understanding. At the same time, there are many misrepresentations and unprofessional expressions in the paper

(2)    In the INTRODUCTION section, the Figure 1 should be deleted. This paper is a research paper, not a review. Also, literature review of this paper was not thorough. Please re-write the section of Introduction with mentioning other researchers’ works through comparison and/or discussion based on a state-of-art literature review. For example, the review of the laser drilling of structural ceramic (doi.org/10.3390/nano12020230), CW drilling of carbon/carbon composites (2023, 10.1016/j.ijmachtools.2022.103978), combined pulse laser drilling method (10.1016/j.optlastec.2022.108209, 10.1016/j.optlastec.2022.109053).

(3)    In the Materials and Methods section, why a f-θ mirror with a focal length of 163 mm is introduced? In addition, Figure 8 should be revised to this part.

(4)    In the Figure 4, the ablation width showed in the Fig. 4(b) is less than 4 um, but the radio of the focused femtosecond laser is around 3 um?

(5)    In the line 191, a scanning space interval of 10 um is hard for me to understand. Also, why choose 10 um since the radio of the focused femtosecond laser is around 3 um?

(6)    How is the HAZ is determined by the upper and/or lower surface of the hole? We suggest that cross-section of the holes should be provided to give more information of the drilling quality, especially for this kind of multi-layer material. Also, there is no SEM figure in the paper.

(7)    The data in the paper are single points and do not conform to the principle of multiple measurements in the experiment, and need to be supplemented with more experimental data, such as adding the statistical error.

(8)    Figure 13 is hard for me to understand.

(9)    How about the drilling efficiency? Also, a taper angle of 5° is not a very high quality, why can't it be improved. What are the reasons for the formation of taper angle and how to go about regulating it?

(10)  In my opinion, the heat accumulation in femtosecond laser drilling is not an important concern of the paper, as all papers on femtosecond drilling consider this point, so the paper needs to further elaborate on its own innovation points.

Author Response

(The authors gave the same response as above.)

Reviewer 3 Report

The paper is devoted to femtosecond laser drilling of submillimeters holes in carbon fiber reinforced polymer. The features of the parameters of the formed holes under different modes of laser ablation are investigated. Also, that very importsnt for laser processing, the heat affected zone is minimized by selecting the scanning trajectory, speed and energy of the laser pulse. The results have applied significance and can be used in technological applications.

There are comments on the work:

1. It's necessary to plot confidence intervals on all experimental graphs. Since this material has a complex structure and it's unclear what kind of variation of values we can observe.

2. It's necessary to explain Figure 5 d, what does the coefficient a and b mean?

3. It's necessary to explain Figure 11 a, why does HAZ change so dramatically from the scanning speed? (at a speed of 0.5 mm/s). And it should be explained how the width of HAZ was determined. It is not clear from the text of the paper.

4. A laser operating with a low pulse frequency was used in the work. To date, the repetiton rate of femtosecond lasers is more than MHz. Are there any advantages to the operation mode at 1 kHz? It's necessary to take this aspect into account in the work. Probably a higher frequency will increase the drilling speed.

After the revision the paper can be published.

Author Response

(The authors gave the same response as above.)

Round 2

Reviewer 1 Report

Dear authors

Thank you for considering all comments, carefully. The manuscript can be published.

Sincerely

Reviewer 3 Report

All comments have been corrected. The paper can be published.